# The Role of Insulin-like Growth Factor-1 (IGF-1) in the Control of Neuroendocrine Regulation of Growth

**DOI:** 10.3390/cells10102664

**Published:** 2021-10-05

**Authors:** Sarmed Al-Samerria, Sally Radovick

**Affiliations:** Department of Pediatrics, Rutgers—Robert Wood Johnson Medical School, New Brunswick, NJ 08901, USA; sr1123@rwjms.rutgers.edu

**Keywords:** IGF-1 singalling, growth hormone, GHRH, energy expenditure, adipose tissues physiology

## Abstract

In mammals, the neuroendocrine system, which includes the communication between the hypothalamus and the pituitary, plays a major role in controlling body growth and cellular metabolism. GH produced from the pituitary somatotroph is considered the master regulator of somatic development and involved, directly and indirectly, in carbohydrate and lipid metabolism via complex, yet well-defined, signaling pathways. GH production from the pituitary gland is primarily regulated by the counter-regulatory effects of the hypothalamic GHRH and SST hormones. The role of IGF-1 feedback regulation in GH production has been demonstrated by pharmacologic interventions and in genetically modified mouse models. In the present review, we discuss the role of IGF-1 in the regulation of the GH-axis as it controls somatic growth and metabolic homeostasis. We present genetically modified mouse models that maintain the integrity of the GH/GHRH-axis with the single exception of IGF-1 receptor (IGF-1R) deficiency in the hypothalamic GHRH neurons and somatotroph that reveals a novel mechanism controlling adipose tissues physiology and energy expenditure.

## 1. Introduction

The neuroendocrine system (NES) is composed of a mixture of specialized cells, which are mainly neuro-peptidergic neurons, located in four hypothalamic nuclei, and capable of secreting neurohormones directly into the bloodstream through the hypophyseal portal blood system [1]. The NES in mammals plays a major role in regulating body growth and reproduction as well as metabolic activity. The hypothalamus, located in the lower region of the diencephalon, is considered the primary source for regulation of the axis producing neural hormones targeting pituitary cells to support multiple biological and physiological activities [2]. Growth Hormone (GH) is a master regulator hormone produced in somatotroph cells and plays a major role in somatic development. The counter-regulatory effects of hypothalamic growth hormone-releasing hormone (GHRH) and somatostatin (SST) primarily regulate GH expression and release, respectively [3]. Additional regulatory mechanisms have been identified, including the peripheral signal, insulin-like growth factor 1 (IGF1), which is the topic of this review.

IGF-1 feedback regulation of GH production has been demonstrated by pharmacologic interventions and in genetically modified mouse models [3,4]. IGF-1 is a polypeptide hormone mainly produced in the hepatocytes and exerts its effect through high-affinity binding to the IGF-1R, located on the cell surface of target tissues [5]. IGF-1 affects a wide variety of biological activities such as somatic cell development, cell differentiation, cortical neuronal activity, regulation of brain development, and is involved, directly and indirectly, in longevity [6,7,8,9,10,11]. Interestingly, IGF-1 has a very potent physiological effect in vivo; however, its effects in vitro are relatively weak unless other hormones or growth factors are present [5]. This mechanism is important to examine and correlate the IGF-1 biological effect in the appropriate tissue and at any specific point in time [5]. Several laboratories, including ours, have identified the key roles of IGF-1 as a major negative regulator of GH production, resulting in a modulation of the growth-related effects of GH [10,12]. Models designed to study IGF-1 modulation of GH synthesis and secretion are associated with a disruption in either downstream signaling or embryologic development of the GH/IGF axis. This review discusses the role of IGF-1 in regulating the GH-axis in somatic growth and metabolic homeostasis. We will present genetically modified mouse models with deletion of the IGF-1 receptor (IGF-1R) in hypothalamic GHRH neurons and somatotrophs that reveal novel mechanisms controlling adipose tissues physiology and energy expenditure.

## 2. The Hypothalamus and Pituitary Gland Axis

The hypothalamic-pituitary axis is a complex, yet, well-defined entity that integrates neuronal and hormonal signals to maintain mammalian growth and somatic development [13]. The hypothalamus is a key regulatory tissue integrating the nervous and the endocrine system to support biological and physiological activities that include reproduction, somatic development, energy balance, and metabolic homeostasis [14,15]. The hypothalamus is strategically located in the lower part of the diencephalon of the brain receiving differentiating signals from other brain areas and, as a consequence, is responsive to environmental signals [14,15]. The hypothalamus communicates with the pituitary gland through two main pathways. First, the neurosecretory cells synthesize hormones, such as oxytocin (OT) and vasopressin or antidiuretic hormone (ADH), that are transported directly to the posterior pituitary gland by axons. Hormones that control the anterior pituitary gland are synthesized and stored in the neuroendocrine cells in the hypothalamus and transported to the anterior lobe through the hypophyseal portal system [7]. The pituitary gland, located at the base of the brain in the sella turcica, is connected to the hypothalamus by the pituitary stalk (infundibulum) [15]. The pituitary gland has two main regions, the anterior pituitary, and the posterior pituitary, responsible for synthesizing nine hormones that govern essential physiological activities.

### 2.1. The Anterior Pituitary

The anterior pituitary also referred to as the adenohypophysis, originates from the oral ectoderm during embryonic development [16]. It is enclosed by a network of blood capillaries originating from the hypothalamus, as a part of the hypophyseal portal system, responsible for transporting hormones from the hypothalamus to the anterior pituitary and from the anterior pituitary to the circulatory system. Hence, the hypophyseal portal system prevents hypothalamic hormones from entering directly into the circulation. [13]. The seven hormones produced from the anterior pituitary gland: GH, prolactin (PRL), thyroid-stimulating hormone (TSH), melanin-stimulating hormones (MSH), adrenocorticotropic hormone (ACTH), follicle-stimulating hormone (FSH), and luteinizing hormone (LH) [16]. The hormones produced from the anterior pituitary are referred to as trophic hormones because they exert their biological activities on the other endocrine tissues. Anterior pituitary hormone production is tightly controlled by the regulatory hormones produced from the hypothalamus, which may be stimulatory or inhibitory [3,17,18].

### 2.2. The Posterior Pituitary

The posterior pituitary lobe originates from neuro-epithelia cells and is therefore referred to as the neurohypophysis. It is anatomically and structurally differentiated from the anterior lobe of the pituitary gland [19]. The posterior lobe consists of neuro-glial cells and nerve fibers extending from the hypothalamus and is considered an extension of the brain [13]. The two hormones secreted by the posterior lobe of the pituitary gland, OT and ADH, are produced by neurosecretory cells in the hypothalamus and transported through the cell axons to be stored in the posterior lobe, from which they are secreted into the circulation system by neuronal signals from the hypothalamus [19].

## 3. IGF-1 and the IGF-1 Receptor

In 1978 Rinderknecht and colleagues at the University of Zurich isolated circulating factors with insulin-like activities, which could be distinguished from insulin by their lack of cross-reactivity with insulin antibodies. Their growth-promoting activity was demonstrated when chemically defined media was supplemented with these factors at low concentrations in vitro. These substances were termed IGF-1 and 2 based on their structural homology with insulin [20]. The same group provided the primary structure and the amino acid sequences of the IGFs.

IGF-1 is a polypeptide hormone with high structural homology with insulin and binds with high affinity to the IGF-1R, activating both the mitogen-activated protein (MAP) kinase and phosphoinositide 3-kinases PI3K signaling pathways in target tissue [6,21]. IGF-1 is mainly produced from liver hepatocytes, and its production and release are primarily controlled by GH [5]. IGF-1 is also expressed in nearly every tissue in the body and plays a pivotal role in regulating a wide variety of bioactivities such as cell proliferation, differentiation, and survival [6,7]. GH/IGF-1 levels dramatically decrease with age, suggesting that a reduction in IGF-1 biological activity is associated with age-related changes to the organism [7].

Using multiple experimental methodologies, including in vivo and in vitro models, IGF-1 has been shown to possesses potent bioactivity to induce cell growth and differentiation of targeted tissues [5]. Despite the similarity between IGF-1 and insulin, insulin plays a major in regulating short-term anabolic activities such as mediating glucose homeostasis and lipid and protein synthesis, while IGF-1 primarily mediates long-term action including cell fate and survival [5]. IGF-1 exerts it is biological activities by binding to the IGF-1R on target tissues [18]. The IGF-1R is a tetrameric glycoprotein-tyrosine kinase receptor, consisting of two extracellular α subunits and two intracellular β subunits that facilitate downstream signals transduction [22,23]. The binding of the IGF-1 ligand to the receptor on the cell surface leads to the activation of two major pathways (MAP) kinase and the PI3 kinase to regulate the IGF-1 response on target tissues [24,25]. In addition, several isoforms of IGF-1 bind to acid-labile subunits (ALS) to mediate ligand/receptor complex formation [26]. IGF-1 has a very short half-life. Therefore, its biological activities are regulated in a spatiotemporal manner to control IGF- 1/IGF-1R levels in the circulation [27,28,29]. Insulin-like growth factor-binding proteins (IGF-1BPs), described initially as free serum carriers, are abundantly expressed in most tissues and play a major role in mediating the biological activities of IGF-1 through autocrine/paracrine modes of action [27]. IGF-1BPs have been shown to inhibit the action of IGF-1. However, several recent studies have demonstrated an up-regulatory mode of action by unclear mechanisms [27,28]. Despite the high structural homology of IGF-1 with insulin, the IGF-1BPs bind exclusively to IGF-1 [27]. Recently, several members of the IGF-1BP family have been shown to regulate other physiological activities in an IGF-independent mechanism including, interaction with other proteins in the extracellular and intracellular space, and mediate the interactions of other growth factor pathways such as transforming growth factor-beta (TGFβ) and epidermal growth factor (EGF) [27]. In humans, more than 99 % of circulating IGF-1 is found to be combined with IGF-1BPs with a relatively prolonged half-life (15 h) compared to unbounded IGFs (10–12 min) [30,31].

A prior study in rodents has shown that food restriction during the early postnatal period (lactation) caused permanent growth retardation and later metabolic changes correlated with lower serum IGF-1 levels compared to the normally fed pups [32]. In the normally fed pups, IGF-1 preferentially stimulates GHRH-neurons growth through two main pathways, PI3K/AKT and ERK/MEK, with a higher contribution of the PI3K/AKT pathway [33]. GHRH-neurons harvested from underfed pups showed a reduction in the GHRH growth, inhibition of axon elongation, which causes lower innervation of the median eminence by the GHRH axon and becomes insensitive to the growth-promoting effects of IGF-1 compared to the age-matched normally fed pups. This loss of function does not involve changes in IGF-1R and ERK/MEK rather is caused by a defect in the AKT activation pathway [33]. IGF-1 is synthesized and produced by almost all tissues and plays a fundamental role in cell differentiation, cell growth, and development [34,35]. In vivo studies using cell-specific *Igf-1* gene knockout mice showed that almost 75% of circulating IGF-1 is produced by the liver, which is responsive to somatotropic GH [36,37]. GH binding to the hepatic GH receptor (GHR) stimulates the production and release of IGF-1 peptides into the circulation [36,38]. IGF-1 exerts its biological effects by binding to the IGF-1R on target tissues [35]. The bioavailability and physiological effects of IGF-1 are regulated by a group of secreted proteins known as IGF-1BPs, which bind with high affinity to IGF-1 to act as transport proteins for circulating IGF-1 [39]. The studies using cell-specific *Igf-1* gene knockout mice have demonstrated that locally produced IGF-1 is more effective than systemic IGF-1 in the control of various biological activities, including somatic cell development, cell differentiation, central nervous system (CNS) development, and embryonic development [6,36,40,41]. In addition to the liver, many other organs and tissues produce IGF-1. These non-hepatic derived, autocrine and paracrine forms of IGF-1 bind to IGFBPs with lower affinity than hepatic IGF-1. 

## 4. IGF-1 and IGF-1R Expression in Neuroendocrine Tissues

In rodents, mRNA expression of IGF-1, IGF-2, and IGF-1R was found during early embryonic development and in the adult by in situ hybridization. The IGF-1R gene has a uniform, stable pattern of expression and distribution in all neuroepithelial cell lineages [42]. High levels of IGF-1R and IGF-1 gene expression were observed in the sensory and cerebellar projection of neurons during late postnatal development [42]. In the cerebral cortex and during hippocampal formation, IGF-1 and the IGF-1R are present in specific cell populations; IGF-1R mRNA is highly expressed in the pyramidal cells in Ammon’s horn, in granule cells in the dentate gyrus, and pyramidal cells in lamina VI of the cerebral cortex [42]. On the other hand, IGF-1R mRNA is expressed in isolated medium- to large-sized cells randomly distributed throughout the hippocampus and iso-cortex [42]. In addition, the IGF-1R and IGF-2 are highly expressed in the choroid plexus, meninges, and vascular sheaths [42]. In the rat pituitary gland, IGF-1/IGF-1R is expressed in all of the endocrine cells, with the highest levels of protein expression in the corticotrophs, somatotrophs, and gonadotrophs. Low levels of IGF-1R expression are present in the thyrotrophs and lactotrophs [43].

## 5. The Role of IGF-1 in the Hypothalamic-Pituitary-Somatotroph Axis (HPS Axis)

Under normal biological and physiological conditions, the HPS axis is very sensitive and highly regulated to influence somatic growth. GH and IGF-1 have a definitive role in regulating somatic development and are involved, directly and indirectly, in metabolic homeostasis and body growth [44,45]. GH production and release from the pituitary somatotrophs is controlled by hypothalamic GHRH, SST, and the GHRH-R on the pituitary somatotrophs [3,46,47]. The activation of GHRH-R by its ligand, GHRH, stimulates GH secretion into the circulation to exert its biological effects by binding to the GHR [48]. In the liver, the activation of the hepatocyte GHR stimulates the production of IGF-1, as well as IGFBPs and ALS, which are responsible for transporting IGF-1 in the circulation [48,49,50]. To highlight the role of IGF-1 at the hypothalamic level, a study in rodents showed that food restriction during the early postnatal period caused permanent growth retardation and later onset of metabolic changes associated with lower serum IGF-1 levels compared to the pups fed a regular chow diet [32]. Underfed pups had a reduction in GHRH neuronal out-growth with decreased axon elongation into the median eminence, rendering the neuron insensitive to the growth-promoting effects of IGF-1. In the pups fed a regular diet, IGF-1 preferentially stimulated GHRH-neuronal growth through both the PI3K/AKT and ERK/MEK pathways, with a more significant contribution of the PI3K/AKT pathway [33]. IGF-1 signaling in the food-restricted pups resulted in a defect in the AKT activation pathway, but IGF-1R expression or ERK signaling was not affected [33].

The ablation of IGF-1R in the pituitary somatotroph resulted in an increase in *Gh* mRNA expression in the pituitary and a modest increase in serum GH and IGF-1 levels. This observation demonstrated the role of IGF-1 in regulating GH production by negative feedback in the somatotroph [3]. These findings in a transgenic mouse model will be discussed in detail in the next section.

## 6. Transgenic Mouse Models with Alterations in the IGF-1 Signaling System

Using gene-editing technology, several transgenic mouse models have been developed to study the role of IGF-1 in the GH-axis, including overexpression of GHRH, GH gene deletion, overexpression of IGF-1 or the IGF-1R, and IGF-1R deletions (Palmiter et al., 1982, Behringer et al., 1988, Mathews et al., 1988, Liu et al., 1993). These mouse models provided evidence of the critical interplay between IGF-1 and GH in the control of mammalian growth and metabolism. However, some of these models are associated with significant disruption in the GH-axis, limiting insight into the mechanism of IGF-1 regulation of GH production. Moreover, despite the pronounced biological effects of IGF-1 in vivo, the in vitro effects are relatively weak unless studied in combination with other hormones or growth factors (Hakuno and Takahashi, 2018). These findings limit our ability to correlate data generated from in vitro studies with the in vivo role of IGF-1. Therefore, this review will focus on the in vivo models exclusively.

### 6.1. Somatotroph IGF-1 Receptor Knockout Mouse Model (SIGFRKO)

Using a Cre/lox strategy, our laboratory developed a novel transgenic mouse model that maintained the integrity of the hypothalamic-pituitary GH-axis, with the single exception of somatotroph-specific IGF-IR deletion, termed somatotroph IGF-1R knockout (SIGFRKO) [3]. The ablation of the IGF-1R in the SIGFRKO mouse model resulted in an increase in Gh mRNA expression in the pituitary and a modest increase in serum GH and IGF-1 levels. This study demonstrated the role of IGF-1 negative feedback in regulating GH production at the level of the somatotroph. Furthermore, Ghrh and Sst mRNA gene expression suggested that compensation at the level of the hypothalamus prevented the dramatic effects on somatic growth observed in other mice models [3]. Interestingly, SIGFRKO mice had a normal linear growth trajectory, however, at 14 weeks of age began to experience a decline in the velocity of weight gain compared to controls mice. SIGFRKO mice had significantly higher energy expenditure, higher VO_2_, lower VCO_2_, and less fat mass and percentage body fat with no change in lean muscle mass. In addition, the calculated respiratory exchange ratio (RER) was significantly decreased in the SIGFRKO mice compared to the controls [51]. Histological examination of the fat depots confirmed a decreased size of the adipocytes in SIGFRKO mice [51]. This mouse model suggested an additional regulatory role of the IGF-1 in the hypothalamus, which required additional studies.

### 6.2. Mouse Models Deleting the IGF-1R from GHRH Neurons (GIGFRKO) and the Pituitary Somatotrophs and GHRH Neurons (S-GIGFRKO)

Two novel transgenic mouse models have been developed to provide further insight into the mechanism of IGF-1R feedback in GHRH neurons and somatotrophs. One has a deletion of the IGF-IR in GHRH neurons, termed GIGFRKO, and the other a deletion of the IGF-1R in both GHRH neurons and somatotrophs, termed S-GIGFRKO [52]. Both mouse models had normal linear growth, but at 14 weeks of age, males and females displayed a reduction in body weight compared to their age and sex-matched controls. Indirect calorimetry demonstrated a higher O_2_ consumption associated with an increase in energy expenditure. This was not associated with either food intake or total activity. Adipocytes from the experimental mice were smaller compared to the controls. These transgenic mouse models provide additional confirmation of the combinatorial role of the IGF-1 signaling system in regulating GH production and highlight a new IGF-1R-GHRH-GH-mediated pathway to regulate GH synthesis and secretion. The effects of IGF-1 on GH gene expression and serum GH levels are direct and indirect, as demonstrated by the changes in Ghrh and Sst mRNA in the hypothalamus, concurrent with changes in the GHRH-R gene expression in the pituitary. Considering the increased interest in using GH as a therapeutic agent to overcome obesity, the knowledge generated from these studies may have significant translational implications. These studies are limited as the complex interactions between fat deposition and lipolytic activity may not alone be explained by GH elevation, since the mice also had slightly increased IGF-1 levels. However, these novel mouse models provide a powerful system not only for demonstrating the functional role of IGF-1 in the somatotroph and the hypothalamus but also highlight an IGF-1R-GHRH-mediated pathway for regulating body weight and energy balance [52].

## 7. Transgenic Mouse Models with Altered GH Expression

Several mouse models have been developed to study the role of IGF-1 on the GH-axis using gene-editing technology. These transgenic mouse models provided evidence of the critical interplay between IGF-1 and GH in the control of mammalian growth and metabolism.

### 7.1. GH ^−/−^ Mouse Model

In 2019, List et al. created a mouse model characterized by the targeted ablation of the GH gene (GH^−/−^) [53]. The GH^−/−^ mice are approximately 50% of the size of wild-type littermates. Circulating serum GH was significantly decreased and IGF-1 levels were undetectable in males and females. The GH^−/−^ mice were also insulin sensitive but glucose-intolerant associated with a significant reduction in pancreatic islet size. The GH^−/−^ mice were responsive to GH treatment, making them an excellent model to study GH replacement therapy.

### 7.2. GHR^−/−^ Mouse Model

The first transgenic mouse model with total body ablation of the GHR (GHR^−/−^) was developed in the Kopchick laboratory using a homologous gene targeting strategy [54]. Similar to the GH^−/−^ transgenic mouse model, the deletion of GHR was associated with severe postnatal growth retardation. The mice had a significant elevation in circulating GH levels, a dramatic reduction in serum IGF-1 level, and were completely insensitive to GH [54,55]. The majority of body organs were decreased in size when compared to wild-type littermates. However, no change was observed in the size of the brain in the GHR^−/−^ mice. This observation suggested that brain growth and development are less dependent on the biological actions of GH [56,57]. The GHR^−/−^ mice were obese mainly due to increased subcutaneous white adipose tissue. In addition, the GHR^−/−^ mice are highly insulin-sensitive and glucose-intolerant associated with fewer and smaller pancreatic cells [58]. Most interestingly, the GHR^−/−^ mice hold the Methuselah mouse prize for “the world’s longest-lived laboratory mouse [59]. The GHR^−/−^ has proved to be an important tool in elucidating multiple aspects of GH activity.

### 7.3. Mouse Model Overexpressing GH

The transgenic mice overexpressing bovine GH (the giant bGH) were obese, had increased food intake, but less percentage body fat than the wild-type littermate controls. Furthermore, these transgenic mice were hyperinsulinemic and displayed impaired gluconeogenesis. The serum IGF-1 levels were increased by 90% compared to the control littermates, and IGF-1 mRNA was increased in subcutaneous, epididymal, retroperitoneal white adipose tissues (WAT), and brown adipose tissue (BAT) depots.

### 7.4. Mouse Models of Altered IGF-1 Signaling

The somatomedin hypothesis formulated in 1972 states that liver-derived IGF-1 plays a key role in GH production and consequently regulates postnatal growth and development [60]. IGF-1 exerts its biological activity through high-affinity binding to the IGF-1R in targeted cells in an autocrine/paracrine and endocrine manner [61,62]. Several transgenic mouse models have been developed to study the role of IGF-1 in the GH-axis.

### 7.5. Mouse Model with a Whole-Body Deletion of IGF-1 and the IGF-1R

The first mouse model of a total-body deletion of the IGF-1 gene (IGF-1^−/−^) was reported by Liu et al. in 1993 [23]. This mouse model demonstrated the crucial role of IGF-1 in regulating prenatal and postnatal body growth and development. The total deletion IGF-1 was associated with a high rate of neonatal death and the surviving pups had severe growth retardation. Mice with a deletion of the IGF-1r gene (IGF-1R^−/−^) died at birth due to severe respiratory failure and displayed severe growth deficiency [23]. Because the liver is believed to be the major source of circulating IGF-1, Yakar et al. developed a unique mouse model with deletion of the IGF-1 gene in the liver and termed it Liver IGF-1 knockout (Liv-IGF-1-KO). This model was designed to assess the importance of circulating (endocrine IGF-1) vs. autocrine/paracrine roles of IGF-1 in somatic growth [63]. The deletion of IGF-1 in the liver resulted in a significant reduction in the circulating levels of IGF-1 in the fetus and during the early postnatal period, followed by a steady increase during puberty. The reduction in serum IGF-1 levels was associated with a significant increase in serum GH levels, likely due to inhibition of the negative feedback at the level of the hypothalamus and/or pituitary (see above SIGFRKO and GIGFRKO). Igf-1 mRNA was not present in the liver of Liv-IGF-1-KO mice. However, Igf-1 mRNA levels in the spleen, heart, fat, muscles, and fat were not affected. Interestingly, the lengths, body weights, and femoral lengths of the Liv-IGF-1-KO mice were similar to the wild-type littermates. The wet weight of the liver in the Li-IGF-1-KO mice was significantly higher than controls, but there were no differences in the weight of other major organs, including the heart and kidney. In addition, the IGF-1-KO mice were fertile and gave birth to litters of normal size. These findings suggested that circulating IGF-1 has a limited role in somatic growth and development and that the majority of growth-promoting activities are mediated by the locally produced IGF-1. This model also confirmed that the liver is the major contributor to the pool of circulating IGF-1 [63].

### 7.6. Brain-Specific IGF-1 R^−/+^ Knockout Mouse Model

In mammals, somatic growth and development involve common major hormonal pathways regulated by the neuroendocrine system [64]. Data generated from invertebrate experimental models suggest that alteration in the IGF-1 signaling pathway in the CNS that decreases IGF-1 and GH levels limits somatic growth and development and prolongs life span [65,66]. To study the role of IGF-1 signaling in the CNS, Kappler, et al., using a conditional mutagenesis technique, developed a transgenic mouse model, bIGF1RKO, characterized by conditional ablation of IGF-1R from the brain [67]. Homozygous deletion of IGF-1R in the brain (bIGF1RKO ^−/−^) resulted in severe growth retardation, elevation plasma IGF-1 levels, microcephaly, infertility, and abnormal behavior. Furthermore, the bIGF1RKO ^−/−^ mice had a shorter life span than the heterozygous mutant (bIGF1RKO ^−/+^) and the control wild-type. Therefore, the homozygous mutant was not considered a suitable model for studying healthy longevity. The heterozygous mutant (bIGF1RKO ^−/+^) was healthy and exhibited normal behavior. Early postnatal body growth of the bIGF1RKO ^−/+^ mice was normal, however, growth retardation became evident at 20 days of age. At 12 weeks of age, bIGF1RKO ^−/+^ mice were shorter and weighed 90% less than the control mice. GH secretion was significantly reduced and no changes were observed in IGF-1 levels throughout development.

## 8. The Role of the IGF-1 Signaling System in Glucose Metabolism

IGF-1 has been shown to bind to the insulin receptor, but with lower affinity than to insulin. The structural similarity between IGF-1, insulin, and their receptors allows for converging physiological and biological effects. While insulin plays a major role in regulating short-term anabolic activities such as glucose homeostasis and lipid and protein synthesis, IGF-1 primarily mediates longer-term actions that include cell fate, survival, and glucose homeostasis [5,68]. IGF-1 has been shown to modulate glucose transport in fat and muscle, inhibit liver glucose output, modulate hepatic glucose production (HGP), and lower blood glucose while suppressing insulin production [69,70].

IGF-1 binds to both the IGF-1R and the insulin receptor (IR) during physiological homeostasis, to form the IGF-1/insulin receptor complex [71]. This complex includes one alpha and one beta subunit from the IR and one alpha and one beta subunit from the IGF-1R. The hybrid receptor complex exhibits a 20-fold higher binding affinity to IGF-1 than insulin and has a crucial role in modulating insulin receptor-linked signaling activities such as tyrosine kinase phosphorylation and glycogen synthesis [72]. These observations suggest that the physiological concentration of IGF-1 may have a role in stimulating insulin-like actions. An in vitro study using rat skeletal muscle revealed that exogenous administration of IGF-1 to the cell culture increased glycogen synthesis and glucose transport and utilization independent of insulin [73].

An in vivo study using a transgenic mouse model characterized by a dominant-negative IGF-1R specifically targeted the skeletal muscle (KR-IGF-1R) demonstrated glucose intolerance at 8 weeks of age and overt diabetes at 12 weeks of age [74]. The expression of the KR-IGF-1R resulted in the formation of an inactive form of the hybrid receptor, thereby impairing its function. Furthermore, the study provided evidence that the KR-IGF-1R mice had impaired pancreatic cell development at a relatively early age, explaining their diabetes at 12 weeks of age.

A study by Yakar et al. using the liver IGF-1 deficient mouse model (LID) demonstrated that the reduction in circulating IGF-1 correlated with a fourfold elevation in serum insulin levels and impaired glucose clearance. These data suggested that insulin resistance was caused by the reduction in circulating IGF-1 in the LID mice. The administration of recombinant human IGF-1 to the LID mice resulted in restoring the glucose response to an acute injection of insulin. Thus, these data generated in LID mice demonstrate that a normal circulating IGF-1 level is required for normal insulin sensitivity [63]. Previous studies demonstrated that mice were given IGF-1 by intracerebroventricular (ICV) injection or by CNS delivery of an Adeno Associated virus 2 (AAV2) encoding IGF-1 had improved insulin sensitivity and glucose tolerance, decreased Pomc levels in the hypothalamus, and increased uncoupling protein 1 (UCP-1) expression in BAT tissues [75].

## 9. The Role of the IGF-1 Signaling System in Obesity

In 1997, the world health organization (WHO) announced that obesity and its associated metabolic complications are a global epidemic and a major public health challenge. The incidence of obesity has risen sharply in the last four decades, such that if this trend continues, by 2030, the majority of the world’s adult population will be overweight or obese [76]. Previous studies have shown that obesity is accompanied by numerous pathological abnormalities such as dyslipidemia, high hypertension, increased insulin secretion, leading to insulin resistance, type 2 diabetes, and cardiovascular diseases [21,77].

Adipocytes are the main structural unit of the adipose tissue and play essential roles in multiple physiological and pathophysiological conditions [78]. Adipocytes are the only cells capable of storing energy and can detect and respond to changes in systematic energy balance [79]. An in vitro study using human mesenchymal stem cells (HMSCs) demonstrated that IGF-1, at low concentrations, was directly involved in preadipocyte differentiation, clonal expansion, lipid droplet formation, and growth [80]. This study also confirmed that the IGF-1R was predominantly expressed in the preadipocytes, whereas it was not detected in mature adipocytes [81]. Although the IGF-1R was abundantly expressed in the preadipocytes, IR was undetectable, suggesting that the differentiating effects of IGF-1 and insulin were mediated solely by the IGF-1R. [80].

Several transgenic animal models in which IGF-1 signaling has been altered in adipose tissue demonstrated that IGF-1 is indirectly involved in mediating lipid synthesis and lipolysis activities by modulating GH and insulin lipolytic activities. Another study in a transgenic mouse model characterized by inactivation of the IGF-1R in the adipose tissue (*IGF-1R-aP2Cre*) demonstrated that IGF-1R signaling in adipocytes does not appear to play an important role in adipocyte development in vivo. The *IGF-1R-aP2Cre* mice exhibited a modest increase in adipose tissue mass correlated with increased lipid accumulation in the epi-gonadal fat pad. The circulating IGF-1 level in *IGF-1R-aP2Cre* mice was elevated and associated with an increase in the trajectory of somatic growth. *IGF-1R-aP2Cre* mice had an increase in IGF-1 mRNA in the liver and adipose tissue. Interestingly, the administration of exogenous recombinant IGF-1 to adipocyte cell cultures extracted from the *IGF-1R-aP2Cre* mice resulted in a significant increase in IGF-1 mRNA whereas, the opposite effect was noted in the wild type adipocytes. These observations led to the conclusion that the IGF-1R in the adipocyte regulates IGF-1 gene expression through a negative feedback mechanism, leading to an increase of circulating IGF-1 to regulate somatic growth [82]. This transgenic mouse model was reported to have limitations as a previous study showed that the aP2 promoter had compromised recombination efficiency [83].

In 2016, the Kahn laboratory developed a novel transgenic mouse model lacking the IGF-1R in adipose tissue (F-IGFRKO) using the Cre-recombinase transgene driven by the adiponectin promoter, which was shown to be more adipocyte-specific than the previous model.

Deleting the IGF-1R in adipose tissue resulted in a reduction in WAT and BAT (~25%), whereas mice lacking both IGF-1R and IR showed a complete lack of WAT and BAT. Despite the reduction in BAT mass in F-IGFRKO mice, the thermogenic activities mediated by BAT were not affected. These results indicated that IGF-1 is essential for normal WAT and BAT development, however, it is not crucial for maintaining normal thermogenic activities.

## 10. The Role of the IGF-1 Signaling System in Modulating Aging and Longevity

The aging process is defined as gradual deterioration in the biological and physiological functions concurrent with continual adaptation to metabolic changes [6]. Several theories have been postulated to elucidate the process of aging, however, the exact mechanisms are not well understood [84]. The programmed aging theory proposed that gradual depletion of systematic neuroendocrine function can severely impact the aging process. The endocrine-related theory states that synchronization of the biological clock of aging is mediated by hormone production and release to control the aging process and regulates adaptation to metabolic changes. The neuroendocrine system plays a major role in regulating several physiological activities, including cell interaction, somatic growth, and maintaining metabolic homeostasis [85]. Accumulating evidence generated from in vivo studies in mice and worms suggested that inhibition of the GH/IGF-1/insulin signaling system can prolong life span, downregulate the aging process, and prevent (to some extent) age-related diseases such as cancer, cardiovascular and metabolic diseases. [21,86]. Prior studies on laboratory animals and humans demonstrated that GH/IGF-1 level gradually decreases with age suggesting that a reduction in IGF-1 biological activity is associated with age-related changes in the organism [7]. For example, in the roundworm *Caenorhabditis Elegans (C. Elegans)* the IGF-1 signaling pathway consist of many proteins encoded by the daf-2 gene [65]. Mutation in this gene causes a reduction in the IGF-1 signaling pathway and appeared to enhance fertility, activity in adult *C. elgance* hermaphrodites, and expand life span to the double compared to the wildtype [65,66]. Conversely, increases in IGF-1 signaling activity have been reported to be associated with a reduction in lifespan [66]. In the fruit fly *Drosophila melanogaster (D. melanogaster)* a mutation in the insulin receptor substrate CHICO associated with reduction IGF-1 signaling pathway activity reported to prolong life span and protect the mutant fly against certain bacterial infections [87].

Several transgenic mouse models have been developed to understand the impact of IGF-1 signaling on the process of aging and longevity. A typical example is the Ames and the Snell transgenic mice. These two mouse models are characterized by a mutation in the PROP-1 and PIT-1 gene, respectively [88,89]. The proteins encoded by these two genes are involved directly in the pituitary cell differential and development which ultimately leads to the production of GH, prolactin, and Thyroid Stimulating Hormone (TSH) [6,89]. The homozygous mutant mice showed a major reduction in IGF-1 production and lacking of all three hormones [90,91]. These mouse models exhibited significant increases in lifespan (~40–70% increase compared to wildtype). Interestingly, Ames mice have been subjected to a caloric restriction regime showed a further increase in their lifespan [92].

In 2018, Huffman laboratory in Albert Einstein College of Medicine/US conducted a preclinical study using IGF-1R monoclonal antibodies (mAb) on 18 months old mice. Female mice treated with IGF-1 mAb exhibited an increase in median lifespan by 9% with a significant reduction in neoplasm and inflammation markers [93].

## 11. Conclusions

In conclusion, the current review discusses the combinatorial role of IGF-1 signaling in regulating GH production and, for the first time, highlights a new IGF-1R-GHRH- GH-mediated pathway to regulate GH production at the level of the hypothalamus using transgenic mouse models developed in our laboratory. Further, these models provided insight into an IGF-1R-GHRH-mediated pathway for regulating body weight and energy balance. Taking into consideration the increased interest in using IGF-1/GH as a therapeutic agent to overcome obesity, we believe that data generated from the two models presented in this review have significant translational implications.

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
