# Peer review of "The Role of Insulin-like Growth Factor-1 (IGF-1) in the Control of Neuroendocrine Regulation of Growth"

_cells, 2021, doi:10.3390/cells10102664_

Round 1

Reviewer 1 Report

The manuscript (cells-1371912) entitled “The Role of Insulin Like Growth Factor -1 (IGF-1) in the Control of Neuroendocrine Regulation of Growth" is a review by Sarmed I. Al-Samerria, and Sally Radovick. The authors discussed the role of IGF-1 in the regulation of the GH-axis as it controls somatic growth and metabolic homeostasis.

Main comments.

  1. A section to compare insulin and IGF-1 signaling systems appears to be needed and helpful.
  2. Section 5 should be integrated into section 3 as it does not help the goal of the paper as a standalone section.
  3. Section 8.6 was not relevant to the title. Probably can be integrated into a previous section.
  4. Most of the manuscript is used to describe IGF-1 pathway in tissues other than Neuroendocrine system. Please focus on that, but not other tissues and cells.
  5. A conclusion and future perspective section is needed.

Author Response

Author's Reply to the Review Report (Reviewer 1)

Main comments.

Thank you kindly for taking the time to assess our manuscript titled “The role of Insulin-like growth factor -1 (IGF-1) in the neuroendocrine regulation of growth.” All comments and suggestions made by reviewer 1 were taken into consideration during the preparation of the revised manuscript.

1. A section to compare insulin and IGF-1 signaling systems appear to be needed and helpful.

  • We agree and have developed new subsections that compare insulin and IGF-1 signaling systems. The comparison between IGF-1 and insulin has been highlighted in the manuscript in several sections (Page 5 Line 14-18 Page 6 line 1-4, Page 13 Line 21-25) using “Track changes”.

2. Section 5 should be integrated into section 3 as it does not help the goal of the paper as a standalone section.

  • Again, we agree and thus have integrated section 5 (factors controlling IGF-1 and IGF-1 R expression) into section 3 (IGF-1 and the IGF-1 Receptor). These changes have been highlighted in the manuscript using “Track changes”.

3. Section 8.6 was not relevant to the title. Probably can be integrated into a previous section.

  • We appreciate this insightful suggestion and in response, Section 8.6 (Mouse Model with Liver-Specific IGF-1 Deletion) has been integrated into the previous section and the section subtitle has been changed to (Mouse Models with a Whole-Body and Cell-Specific Deletion of IGF-1 and the IGF-1R).

4. Most of the manuscript is used to describe the IGF-1 pathway in tissues other than the Neuroendocrine system. Please focus on that, but not other tissues and cells.

  • After a careful review consideration of the reviewer’s comments, we have revised the manuscript and further, added a figure to distinguish the role of the IGF-1 pathway in the neuroendocrine system.

5. A conclusion and future perspective section are needed

  • We agree and have added a conclusion section that includes future perspectives on Page

Reviewer 2 Report

In this review, Al-Samerria and Radovick make a revision about the role of IGF-1 in the regulation of neuroendocrine system. This review article is focused on the cellular and molecular mechanisms involved in the control of the Growth Hormone secretion by IGF-I and its control on somatic growth and metabolic homeostasis. The authors suggest a novel mechanism controlling adipose tissues physiology and energy generation. The review is well written and clear. I have some comments that could improve this manuscript.

The authors indicate different functions of the IGF-I. About the effect of IGF-I on the Central Nervous System, they only mention the regulation of brain development. In my opinion, the effect of IGF-I on the neuronal activity should be indicated.

Page 2, first line. In the CNS, IGF-I has an important effect on cortical neurons recorded in vitro (see for example Noriega-Prieto et al. 2021).

Page 8. First paragraph of the Section: “9. The Role of the IGF-1 Signaling System in Glucose Metabolism”. The first sentence of this paragraph is repeated in the fifth paragraph of page 3. Please, remove or rewritten.

Page 9, last sentence of the text. The authors end this review indicating that “Accumulating evidence generated from in vivo studies in mice and worms suggested that inhibition of the GH/IGF-1/insulin signaling system can prolong life span”. This is a very controversial point that should be covered in more detail (see Vitale et al. 2019).

Finally, I suggest to include a final section with some Conclusions.

Author Response

Author's Reply to the Review Report (Reviewer 2)

Main comments.

Thank you kindly for taking the time to review our manuscript titled “The role of Insulin-like growth factor -1 (IGF-1) in the control of neuroendocrine regulation of growth.”  

  1. The authors indicate different functions of the IGF-I. About the effect of IGF-I on the Central Nervous System, they only mention the regulation of brain development. In my opinion, the effect of IGF-I on neuronal activity should be indicated.
    • We agree with reviewer 2 that we should have elaborated on this point using a section to highlight the effect of the IGF-1 signaling system on the central nervous system. Therefore, we added a section on page 6 line 24- page 7 line 4 discussing the role of IGF-1 on the GHRH neurons.
  2. Page 2, first line. In the CNS, IGF-I has an important effect on cortical neurons recorded in vitro (see for example Noriega-Prieto et al. 2021).
    • Indeed, the reviewer's suggestion is very important, thus we implemented these changes and cited the suggested references.
  3. Page 8. First paragraph of the Section: “ The Role of the IGF-1 Signaling System in Glucose Metabolism”. The first sentence of this paragraph is repeated in the fifth paragraph of page 3. Please, remove or rewritten.
    • We apologize for the oversight and have removed the sentence from the paragraph.
  4. Page 9, last sentence of the text. The authors end this review indicating that “Accumulating evidence generated from in vivo studies in mice and worms suggested that inhibition of the GH/IGF-1/insulin signaling system can prolong life span”. This is a very controversial point that should be covered in more detail (see Vitale et al. 2019).
    • We appreciate the reviewer's comment and agree that it is a controversial point that should be covered in more detail. We add a new section on pages 16-17 to cover the topic in more detail.
  5. Finally, I suggest including a final section with some Conclusions.
    • We agree with the reviewer that a conclusion section is needed. Therefore a conclusion section has been added to the manuscript on Page

Round 2

Reviewer 1 Report

The authors have addressed my comments. There is no additional comment.